# The Infection Properties of *Trionyx sinensis* Hemorrhagic Syndrome Virus and the Antiviral Effect of Curcumin In Vivo

**DOI:** 10.3390/ani13233665

**Published:** 2023-11-27

**Authors:** Jinbiao Jiao, Jiayun Yao, Feng Lin, Xuemei Yuan, Lei Huang, Jing Chen, Xianqi Peng, Haiqi Zhang, Shengqi Su

**Affiliations:** 1Key Laboratory of Freshwater Fish Reproduction and Development (Ministry of Education), Research Center of Fishery Resources and Environment, College of Fisheries, Southwest University, Chongqing 400715, China; jiaojinb@126.com; 2Agriculture Ministry Key Laboratory of Healthy Freshwater Aquaculture, Key Laboratory of Fish Health and Nutrition of Zhejiang Province, Key Laboratory of Fishery Environment and Aquatic Product Quality and Safety of Huzhou City, Zhejiang Institute of Freshwater Fisheries, Huzhou 313001, China

**Keywords:** *Trionyx sinensis*, TSHSV, curcumin, antiviral, immune regulation

## Abstract

**Simple Summary:**

*Trionyx sinensis* (Chinese soft-shelled turtle) is one of the main economic animals in the aquaculture industry due to its nutritional value and medicinal value. However, the farming industry is facing various challenges, including water quality, stocking density, and viral disease outbreaks. *Trionyx sinensis* hemorrhagic syndrome virus has high lethality to *Trionyx sinensis*, but the detailed pathogenic characteristics of the virus are unclear, and there is no available treatment for the virus. After experiments, we revealed the pathogenic characteristics of the virus at the cellular and tissue levels and found a material for the culture of the virus. Curcumin is a natural herbal extract that has antiviral, antifungal, and antibacterial properties and acts as an immunomodulator. In vivo tests have shown that curcumin is effective in preventing viral infection and improving the immune response. These results demonstrate that curcumin is a promising strategy for preventing viral disease.

**Abstract:**

*Trionyx sinensis* hemorrhagic syndrome virus (TSHSV) is an aquatic arterivirus causing a high mortality rate for *T. sinensis* (Chinese soft-shelled turtle), but the detailed infection properties of TSHSV are unclear, and no effective treatment is available. In this study, cell culture and histopathology were performed to elucidate the infection properties of TSHSV. Furthermore, the anti-TSHSV and immune-enhancing effects of curcumin were evaluated using survival statistics, qPCR, and tissue immunofluorescence. The results demonstrated that TSHSV could proliferate in the spleen cell line of *T. sinensis*, leading to cytopathic effects. TSHSV damaged the livers, kidneys, and lungs, characterized by cell disintegration and hyperemia. Curcumin at 250 mg/kg improved the survival of *T. sinensis*, and significantly reduced the viral load in the spleens, kidneys, and lungs. Moreover, curcumin inhibited the mRNA expression of immune-related genes, *RSAD2*, *IFN-γ*, and *TNF-α* (*p* < 0.05). In conclusion, these results imply that TSHSV is pathogenic to the spleen cell line, liver, spleen, kidney, and lung of *T. sinensis*. Curcumin effectively inhibits TSHSV and modulates the immune function of *T. sinensis*, so it holds promise as a means to prevent TSHSV.

## 1. Introduction

*Trionyx sinensis* is a highly commercial aquaculture species in Asian countries, especially China, Japan, and Korea, due to its high nutritional and pharmacological values. However, intensive farming has led to increasingly frequent disease outbreaks, resulting in huge economic losses. *T. sinensis* hemorrhagic syndrome virus (TSHSV) is a pathogen that causes a disease characterized by multi-organ hemorrhages and high mortality, which has hindered the turtle culture industry [1].

TSHSV is an enveloped single-stranded RNA virus with a diameter of 60–80 nm, belonging to the family Arteriviridae [1,2]. The virus causes hepatomegaly, spotty, and severe congestion in liver tissue, and also apparent intestinal bleeding [3]. In detail, cell necrosis and disintegration were observed in infected livers and intestines [3]. However, a more detailed cytological study of TSHSV has yet to be conducted. The pathogenicity of the lung transcriptome and the liver proteomics have been investigated in preliminary studies. Notably, *RSAD2* was significantly upregulated in mRNA and protein levels after viral infection, suggesting that *RSAD2* was involved in antiviral response in *T. sinensis* [3,4]. Vaccines have achieved outstanding results in preventing aquatic viral diseases. Unfortunately, TSHSV exhibits an antibody-dependent enhancement (ADE), with the antiviral gene *RSDA2* and viral copy highly upregulated after stimulation with polyclonal antibodies [5]. Therefore, alternative prevention options for TSHSV are desperately needed.

Chinese herbal medicine has been extensively applied in the control of aquatic diseases due to its effectiveness, safety, low toxicity, and few side effects [6]. Curcumin, considered as the main active substance of turmeric exerting medicinal activity, possesses antiparasitic, antibacterial, antioxidant, and immunomodulatory properties [7,8]. A curcumin-rich diet enriches B cell populations and postpones the initial proliferation of *Sphaerospora molnari* in the blood [9]. A feed diet containing 150 mg/kg of curcumin was enhanced silver catfish’s resistance to *Streptococcus agalactiae* infection [10]. Curcumin also alleviates the toxicity of acute ammonia in the head kidney macrophage of *Pelteobagrus fulvidraco* by lowering the levels of *NF-κB*, *SOD*, and *ROS* [11]. Notably, curcumin also alleviates in vitro infections caused by several viruses, such as Coxsackie virus [12], hepatitis C virus [13], herpes simplex virus [14], and porcine reproductive and respiratory syndrome virus [15]. However, there are few reports about the protective effect of curcumin on aquatic viral infections. In fish virus research, curcumin suppressed the early stages of viral hemorrhagic septicemia virus (VHSV) infection via rearrangement of the F-actin/G-actin ratio via downregulating HSC71 [16]. Curcumin inhibited Singapore grouper iridovirus (SGIV) by suppressing apoptosis and promoting autophagy [17]. Nevertheless, whether curcumin has a preventive effect on TSHSV is unknown, and requires further clarification.

A cell line sensitive to TSHSV has not been reported, and a prevention strategy is unavailable. Therefore, the permissiveness of cell lines to TSHSV was assessed in this study. Furthermore, the antiviral and immunomodulatory effects of curcumin were evaluated via qPCR and immunofluorescence. Overall, this study may provide an essential basis for the diagnosis and treatment of TSHSV.

## 2. Materials and Methods

### 2.1. Animals and Virus Sample Preparation

All of the healthy *T. sinensis* used in the experiment were obtained from a farm in Huzhou and weighed approximately 100 g each. Prior to the experiment, all turtles were habituated to a fresh water tank at around 30 °C for at least one week, and were confirmed to be free of bacteria, parasites, and viruses. The naturally diseased turtles used for virus preparation were gathered from aquaculture farms in Zhejiang Province, and were identified as TSHSV-positive, following the previous method [1]. For virus preparation, the TSHSV-positive tissues were pooled and homogenized in phosphate-buffered solution (the PBS) (tissue mass (g)/PBS volume (mL) ratio was 1:5), then centrifuged and filtered with a 0.22 μm filter. The obtained crude virus was stored at −80 °C for further experiment.

### 2.2. Electron Microscopic Observation of TSHSV

The two cell lines used were isolated and subcultured based on a method previously reported [18], including the kidney cell line (TSK) and spleen cell line (TSSP) of *T. sinensis*. The two cell lines were cultured in complete L-15 medium (Leibovitz’s L-15 medium, Invitrogen) containing 10% fetal bovine serum (FBS) (Invitrogen, Carlsbad, CA, USA), 100 U/mL penicillin, and 100 μg/mL streptomycin at 28 °C. The monolayer cells were inoculated with crude virus and incubated in a maintenance medium (L-15 medium containing 2% FBS, 100 U/mL penicillin, and 100 μg/mL streptomycin) at 28 °C. When the cytopathic effect (CPE) occurred, cells were collected for virus detection and electron microscope observation. The CPE cells were collected and washed with L-15 medium and PBS, then centrifuged into cell pellets. The cell pellets were processed via a protocol including fixation, dehydration, embedding, sectioning, and staining, as described previously [19]. Electron microscopy was performed in the Center of Cryo-Electron Microscopy (CCEM) at Zhejiang University.

### 2.3. Pathogenic Characteristics of TSHSV

The healthy turtles were randomly divided into six groups (*n* = 30), labeled five TSHSV-infected groups (10^0^, 10^−1^, 10^−2^, 10^−3^, and 10^−4^) and one healthy control group. A series of concentration gradients of the crude virus were used to evaluate virus pathogenicity. The crude virus was serially diluted 10-fold with PBS from 10^0^ to 10^−4^ times. The five virus-positive groups were intraperitoneally injected with 0.2 mL of the corresponding diluted virus, while the healthy control group was injected with PBS. The water temperature was maintained at 30 °C during the infection. The median lethal dose (LD50) of TSHSV was calculated based on 15-day mortality using the Reed–Muench method [20]. In addition, anatomical observation and virus detection were performed on the infected turtles to confirm the TSHSV infection. The turtles were sacrificed after the survival recording for pathological analysis. The livers, spleens, kidneys, and lungs of three infected and three healthy turtles were collected and fixed with 4% paraformaldehyde for more than 24 h. Hematoxylin and eosin staining and microscopic observation of the tissues were performed according to the previous method [21].

### 2.4. Absolute Quantification and Tissue Distribution Characteristics of TSHSV

To measure the viral load, the quantitative primers F1/R1 for TSHSV and the standard fragment primers F2/R2 were designed based on the TSHSV genome sequence (Genbank ID: MH447987) using Primer Premier 5.0 software (http://www.premierbiosoft.com/, accessed on 15 October 2021) (Premier Biosoft, Palo Alto, CA, USA) (Table 1). The polymerase chain reaction (PCR) of primers F2/R2 was performed with 2 × Taq Master Mix (New England Biolabs, Ipswich, MA, USA) according to the instruction manual on a Bio-Rad Thermal Cycler (Bio-Rad, Hercules, CA, USA). The PCR reaction conditions were 94 °C for 3 min, followed by 35 cycles of 94 °C for 30 s, 58 °C for 30 s, and 72 °C for 1 min, and a final extension at 72 °C for 10 min. The purified PCR product was ligated into the PMD-18T vector and cloned into DH-5α competent cells to obtain high-copy plasmid standards. The successful construction of the plasmid was verified via identification of restriction enzyme digestion. Absolute quantification was performed via a ten-fold gradient dilution of the plasmid standard. The qPCR reaction system included 5.0 μL of SYBR Green I Master Mix (Roche, Basel, Switzerland), 3.0 μL of ddH_2_O, 0.5 μL of primers F1 and R1 (10 μM), and 1.0 μL of plasmid standard. The qPCR reactions were performed on a Light Cycler 480 Real-Time PCR system (Roche, Basel, Switzerland), with the following parameters: an initial denaturation at 95 °C for 10 min, followed by a three-step amplification for 35 cycles (denaturation at 95 °C for 10 s, annealing at 60 °C for 10 s, extension at 72 °C for 30 s), and a dissolution curve. Three replications were performed for each concentration, fitting a standard curve between the Ct values and the plasmid copy numbers.

The liver, spleen, kidney, heart, lung, brain, intestine, muscle, and parotid gland of three naturally occurring diseased turtles were collected and used to analyze the tissue distribution of TSHSV. Healthy turtles were intraperitoneally injected with 100LD50 crude virus to clarify the viral replication trends. The top three virus-enriched tissues were collected for virus quantification from one to eight days post-infection (dpi). The viral load was measured as follows: (1) The tissues were ground in liquid nitrogen, and RNA was extracted using RNAiso Plus Reagent (Takara, Shiga, Japan). (2) RNA was reverse transcribed into cDNA using a Prime Script RT Reagent kit (Takara, Shiga, Japan). (3) qPCR reactions were performed as described above. The measured data were converted to viral load using the standard curve.

### 2.5. Comparison of the Protective Effects of Three Chinese Medicines

Three drugs were tested on the turtles, including curcumin, echinacea purpurea polysaccharide, and coumarin-3-carboxylic acid. The healthy turtles were randomly allocated into eight groups (*n* = 50, under the same condition, 30 for survival analysis and 20 for virus quantification). The eight groups included three negative groups, labeled curcumin (CUR), echinacea purpurea polysaccharide (EPP), and coumarin-3-carboxylic acid (CCA), one TSHSV-infected group (TS); three treatment groups (CUR-TS, EPP-TS, CCA-TS); and one healthy control group (CK). The therapeutic drugs were dissolved in saline and administered orally. Specifically, the three negative groups and three treatment groups were administered the dose of 500 mg/kg of corresponding drugs once a day, while the TS and CK groups were fed with an equal volume of saline. The treatment lasted seven days, and drug safety was assessed based on survival. Following the treatment period, the three treatment groups and the TS group were intraperitoneally injected with 100LD50 virus, while the three negative groups and the CK group were injected with an equal volume of PBS. The survival was monitored for seven days. At seven dpi, the spleen, kidney, and lung samples were extracted for virus quantification using the above method (see Section 2.4).

### 2.6. Protective Effects of Curcumin at Different Concentrations

The healthy turtles were randomized into five groups (*n* = 40): three curcumin groups (125 mg/kg, 250 mg/kg, and 500 mg/kg), one TSHSV-infected group (TS, virus challenge with no curcumin treatment), and one healthy control group (CK, neither virus challenge nor curcumin treatment). The procedures of drug dissolution, drug treatment, and viral infection were in accordance with those described above (see Section 2.5). The survival of all groups was recorded up to 14 dpi.

### 2.7. Anti-TSHSV and Immunoregulatory Effect of Curcumin

Two groups were labeled as follows: the curcumin group (CUR, *n* = 60) and the virus group (TS, *n* = 60). The CUR group was fed with curcumin at 250 mg/kg for seven days, as described in Section 2.6, and then infected with the 100LD50 virus. The TS group was fed saline, and all other conditions were consistent with the CUR group. The spleens, kidneys, and lungs were sampled at one, three, five, and seven dpi for virus quantification, as described above (Section 2.4).

The immunofluorescence analysis of TSHSV accumulation in tissues was performed. The livers, spleens, kidneys, hearts, and lungs were sampled at seven dpi, fixed in 4% paraformaldehyde, embedded, and sectioned for fluorescence staining to capture viral enrichment. The antibody A661, specific to TSHSV, was preserved in our laboratory. Detailed procedures were performed in accordance with a previous method [22]. Fluorescence images were captured using a Leica DMi8 fluorescence microscope (Leica, Nussloch, Germany), and the fluorescence intensity was estimated with ImageJ software (version 1.8.0) and expressed as arbitrary units.

The expression of immune-related genes was analyzed at the mRNA level. The samples used for viral load determination were also used for gene expression analysis. The procedures of RNA extraction, reverse transcription, and qPCR were consistent with those described in Section 2.4. The 18S rRNA gene was used as an internal reference gene, and all primers are listed in Table 2. The relative gene expression was calculated using the 2^−ΔΔCt^ method [23].

### 2.8. Data Statistics and Analysis

The data were presented as mean ± standard deviation with 95% confidence interval limits. Data were analyzed with a one-way analysis of variance (ANOVA) and Student’s *t*-test, with *p* < 0.05 as the threshold for significance. Graphs were plotted using GraphPad Prism 8.0 (GraphPad Software, San Diego, CA, USA). Animal survival was analyzed using the Kaplan–Meier method, and significance was determined using the Log-rank (Mantel–Cox) test.

## 3. Results

### 3.1. The Cytopathic Effect Caused by TSHSV Infection

The TSK cells showed no CPE at eight dpi, but cells showed visible shrinkage and massive shedding at ten dpi (Figure 1a). The TSSP cells exhibited shrinkage, roundness, and disintegration at seven dpi (Figure 1b). The TSHSV was detected in both cell lines (Figure 1c). In the CPE TSSP cells, accumulated spherical virus particles with 50–100 nm diameter were observed under a transmission electron microscope (Figure 1d).

### 3.2. Histopathology of Turtles Infected with TSHSV

The TSHSV-infected liver was characterized by swollen cells with vacuolated cytoplasms and condensed nuclei. Confluent lytic necrosis was observed in hepatocytes, which formed a bridging necrosis. Apparent blood stasis was observed in the hepatic sinuses and interlobular arteries. The extensive inflammatory cell infiltration accumulated around the bile duct (Figure 2b). In TSHSV-infected spleen, lymph nodules exhibited severe atrophy, necrosis, and disintegration; lymphocytes were scattered and sparse (Figure 2d). In TSHSV-infected renal tissue, necrotic areas in the renal parenchyma were characterized by distinct hemorrhages. The renal cells exhibited nuclear pyknosis and karyolysis. The renal tubules atrophied and the renal sacs dilated, leading to an enlarged space. Some cells in the glomeruli disintegrated, forming a cavity (Figure 2f). Compared to the normal lung, the TSHSV-infected lung demonstrated apparent structural disorders. The alveolar cells were characterized by nucleolysis and cytoplasmic vacuolation. The elastic fibers in the alveolar walls disintegrated (Figure 2h).

### 3.3. The LD50 of TSHSV

The crude virus was lethal to *T. sinensis*, and TSHSV was detected in the tissues, consistent with the preparation case. The TSHSV tended to be more lethal at higher concentrations (Table 3). The LD50 was 10^−2.114^/0.2 mL (corresponding to 10^6.2^ viral copies), based on the Reed–Muench method.

### 3.4. Absolute Quantification of TSHSV

A 927bp DNA fragment amplified by primers F2/R2 was obtained and cloned into the PMD-18T vector (Figure 3a,b). The standard curve was constructed based on the qPCR reaction of gradient plasmid standards, showing a good linear trend with the following formula: Ct = −3.479lg(viral copies) + 41.429 (Figure 3c). TSHSV was widely distributed in all tissues, with the highest viral load in the spleen, kidney, and lung. There were significant differences between lung and kidney, kidney and liver, liver and muscle, and muscle and brain (*p* < 0.05) (Figure 4a). Overall, the viral loads in the spleen, kidney, and lung tended to increase and subsequently decrease. Notably, the lung reached the highest viral load at two dpi, while the spleen and kidney did so at three dpi (Figure 4b–d).

### 3.5. Anti-TSHSV Effects of the Three Drugs

As demonstrated in Figure 5a, the three virus-negative groups (CUR, EPP, and CCA) all survived within seven days of drug treatment. The results demonstrated the potential protective benefits of all three drugs in enhancing survival. At seven dpi, the survival was 60% in the TS group; 100% in the EPP-TS, CCA-TS, and CK groups; and 90% in the CUR-TS group. Viral assays showed that curcumin diminished viral load significantly in the spleen and lung (*p* < 0.05). Echinacea polysaccharide lowered viral load in the spleen, showing a significant difference (*p* < 0.05). Furthermore, coumarin-3-carboxylic acid had antiviral effects in three tissues, with no significant differences (Figure 5b).

### 3.6. Inhibitory Effect of Curcumin on TSHSV

The survival curves revealed the protective effects of curcumin at different doses. Among curcumin-treated groups, 250 mg/kg demonstrated the highest survival (52.5%), followed by 500 mg/kg (47.5%) and 125 mg/kg (37.5%) (Figure 6a). Considering survival, an optimal dose of 250 mg/kg was chosen for the subsequent tests. Curcumin inhibited the viral load in the spleen significantly at one dpi (*p* < 0.05) (Figure 6b). In the kidney, the relative viral load initially decreased, reaching the lowest level at three dpi (*p* < 0.05), and then increased (Figure 6c). The relative viral load in the lung exhibited an increasing trend, but was significantly lower than in the TSHSV group (*p* < 0.05) (Figure 6d). Immunofluorescence analysis was performed to detect viral protein expression. At seven dpi, curcumin suppressed viral protein levels in the liver, spleen, kidney, and lung, while enhancing viral levels in the heart (Figure 6e).

### 3.7. Effect of Curcumin on the Expression of Immune-Related Genes

The expression of immune-related genes was assayed. The *RSAD2* mRNA expression in the TS group tended to initially increase and subsequently decrease, reaching its highest level at three dpi. Curcumin significantly reduced *RSAD2* mRNA expression in the spleens and kidneys (*p* < 0.05), but significantly enhanced *RSAD2* expression in the lungs (*p* < 0.05). *IFN-γ* mRNA expression in the TS group generally showed a declining trend over time. Except for the kidneys and lungs at three and seven dpi, curcumin significantly reduced *IFN-γ* mRNA expression (*p* < 0.05). Curcumin reduced *TNF-α* mRNA levels in the spleens (0, 1, and 5 dpi), kidneys (0–3 dpi), and lungs (0 and 1 dpi) (*p* < 0.05). However, with the exacerbation of TSHSV infection, the *TNF-α* mRNA level showed an increasing trend in the CUR group, and was even significantly higher than the TS group in the kidney and lung at seven dpi (*p* < 0.05) (Figure 7).

## 4. Discussion

### 4.1. Pathogenic Characteristics of TSHSV

Sensitive cell lines are critical for virus identification, vaccine development, recombinant protein production, and the study of models of pathogenesis and immune mechanisms [24,25,26,27]. TSHSV is a virus derived from *T. sinensis*, but no sensitive cells have been reported. This study found that the spleen cell line infected with TSHSV could exhibit cytopathic effects and replicate the virions. This cell line will provide additional material for diagnosing hemorrhagic syndrome in soft-shelled turtles, identifying viruses, and studying viral biological characteristics. In CPE cells, TSHSV virions were successfully observed in the TSSP, but not in the TSK. The detailed reasons for this need to be further revealed. The virulence and titer of the virus are determined by complex conditions, including cell line, virus isolate, cell density, and incubation temperature [28]. Therefore, the experimental conditions need to be further optimized to improve the replication efficiency of TSHSV. According to a previous report, soft-shelled turtle iridovirus (STIV) multiplied and caused CPE in grass carp ovary (CO), fathead minnow (FHM), grass carp kidney (CK), and blue-gill fry (BF-2) cell lines [29]. Therefore, some fish cell lines can also be selected as candidates for culturing viruses derived from *T. sinensis*, including TSHSV.

Histopathological studies are critical for identifying lesions and providing data for disease diagnosis and treatment. The histopathological features were consistent with a previous study [3]. The hepatocytes were characterized by lytic necrosis and the kidney exhibited atrophied renal tubules and dilated renal sacs. Furthermore, we also found that the lymph nodules exhibited atrophy, necrosis, and disintegration. The alveolar cells displayed cytoplasmic vacuolation and the alveolar walls disintegrated. These features indicated that TSHSV extensively injured the tissues, with the common characteristics of cell necrosis and disintegration. It has been reported that TSHSV has a fatality rate of 100% after one week of infection [1]. However, this rate was 80% after two weeks of infection in this study. Water temperature is a key factor affecting the process of viral infection. Compared with 27 °C, a higher water temperature (31 °C) could prevent the onset of disease and significantly reduce the mortality of shrimp infected with white spot syndrome virus [30]. We found that the water temperature in this study was higher than previously reported (30 °C versus 28 °C) [1]. Therefore, water temperature may be contributing to the difference in lethality. The quantitative PCR revealed evidence that TSHSV could proliferate in all tissues. The top three TSHSV-enriched tissues were the spleen, lung, and kidney, similar to a previous report [1].

### 4.2. Antiviral Effect of Curcumin

The antiviral effects of curcumin, echinacea purpurea polysaccharide, and coumarin have been extensively studied in aquatic diseases [31,32,33]. This study revealed that the three drugs have potential inhibitory effects on TSHSV, especially curcumin. After being treated with 500 mg/kg of curcumin for seven days, the survival was 100%, indicating high safety at the concentration of 500 mg/kg. Therefore, 500 mg/kg was set as the highest concentration for the subsequent concentration gradient test. Curcumin has been extensively studied as an antiviral agent. It plays an antiviral role by blocking the internalization of porcine reproductive and respiratory syndrome virus (PRRSV) and virus-mediated cell fusion [15]. It is worth noting that TSHSV is taxonomically close to PRRSV, both of which belong to the family Arteriviridae. Therefore, the resistance mechanism of curcumin to the two viruses may be similar. The protective effect of curcumin showed tissue variability, with the best effect on the lungs. Curcumin is an effective drug for improving respiratory diseases, alleviating pulmonary fibrosis, and inhibiting the production of lung inflammatory factors to combat chronic lung disease [34,35]. Curcumin also showed temporality, with significant antiviral effects on the spleen and kidney in the early stage, but weak antiviral effects in the later stage. The effect in the lungs also gradually decreased over time, although significantly compared to the TS group. This trend was primarily attributed to curcumin’s rapid metabolism and short half-life in animals [36]. Considering the weakened effect of curcumin as metabolized in *T. sinensis*, continuous or intermittent administration can be attempted in further studies, which may improve the efficacy of curcumin [37]. It is worth noting that simultaneous administration of drugs and virus can achieve efficient antiviral effects in rainbow trout [38], and this reflects the direct antiviral effect of drugs. This may provide a new idea for drug delivery. Curcumin can achieve a sustained-release process when loaded with nanoparticles or liposomes, improving absorption into plasma and the target tissues [39]. These studies will provide new insights into the delivery and efficacy of curcumin.

### 4.3. Regulation of Immune-Related Genes by Curcumin

The level of immune-related genes can reflect the health status and immune response of the organism to some extent [40]. *RSAD2* is one of the genes stimulated by interferon (IFN), and its product viperin can interfere with the proliferation of various viruses through diversified mechanisms [41]. The significantly upregulated protein and mRNA levels of *RSAD2* were noted in TSHSV-infected soft-shelled turtles [3,4]. *RSAD2* was regarded as an anti-TSHSV virus gene and a marker for initiating an immune response. After curcumin treatment for seven days, TSHSV levels and *RSAD2* overall levels showed a downward trend, suggesting that curcumin may play a key role in antiviral activity, compensating for the antiviral response of *RSAD2*.

IFN-γ, a type II IFN, is a cytokine produced by activated T cells and NK cells, and is essential in regulating the antiviral immune response [42]. IFN-γ induces antiviral immunity by modulating the innate immune response and activating adaptive immunity against viruses [43,44,45]. Curcumin has been shown to attenuate liver injury by reducing *IFN-γ* [46]. Similarly, this study also found that curcumin significantly inhibited *IFN-γ* levels in the liver, spleen, and kidney, suggesting that curcumin has beneficial effects on alleviating organ injury and antiviral activity.

TNF-α is a critical pro-inflammatory factor mediating the inflammatory response. Stimulated by invasive pathogens or mechanical injury, *TNF*-*α* may be upregulated to trigger inflammatory responses and damage cells and tissues [47]. Curcumin significantly reduces *TNF-α* levels during the early stages of infection, especially in the spleen and lungs. This was similar to the previous studies in other aquatic animals, and was consistent with the recognized anti-inflammatory function of curcumin [48,49]. However, in the later stages, the inflammatory gene *TNF-α* in the CUR group showed an uptick, similar to the viral trend with a delayed effect. These changes may be due to dilution, degradation, or accelerated metabolism of curcumin in *T. sinensis*, thus reducing the anti-inflammatory response [50]. Therefore, it is critical to study how to improve the efficacy and duration of curcumin.

## 5. Conclusions

This study set out to gain a better understanding of the infection properties of TSHSV and the antiviral effects of curcumin. Based on the above evidence, the splenic cell line (TSSP) was screened for multiplication of TSHSV. TSHSV causes cell disintegration and hyperemia in the livers, kidneys, and lungs. Curcumin at 250 mg/kg exerts a potent anti-TSHSV effect in *T. sinensis*. Curcumin also regulates the levels of two antiviral genes and a pro-inflammatory factor. In summary, curcumin may be a promising dietary supplement for improving disease resistance. Further understanding of the molecular mechanisms affected by curcumin therapy will provide a basis for agents in preventing TSHSV.

## Figures and Tables

**Figure 1 animals-13-03665-f001:**
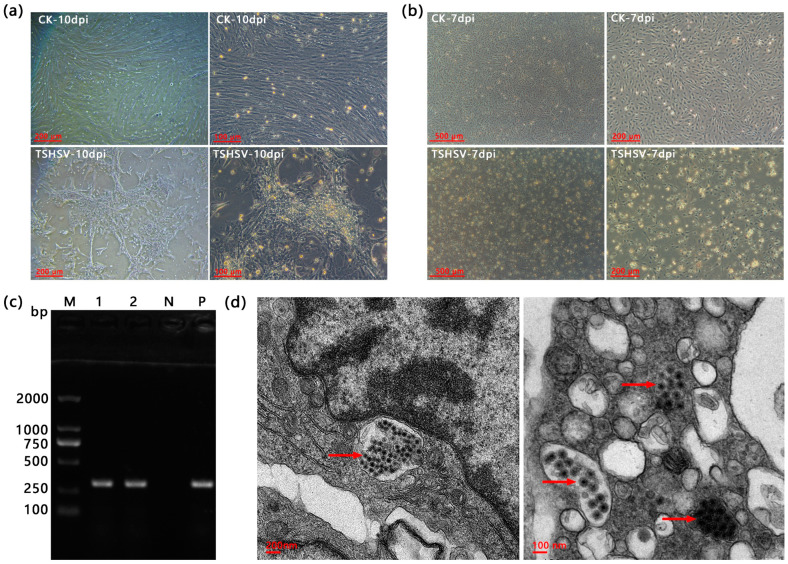
(**a**) The cytopathic effects of TSK infected at ten dpi. (**b**) The cytopathic effects of TSSP at seven dpi. (**c**) The PCR detection results of TSHSV in TSK and TSSP. M: DL2000 Marker, 1: TSK, 2: TSSP, N: negative control (dd H_2_O), P: positive control. (**d**) The transmission electron microscopy (TEM) of splenic cells for virus localization. The red arrows point to TSHSV virions. Note: (**a**,**b**) CK is the normal cells, and TSHSV is the infected cells.

**Figure 2 animals-13-03665-f002:**
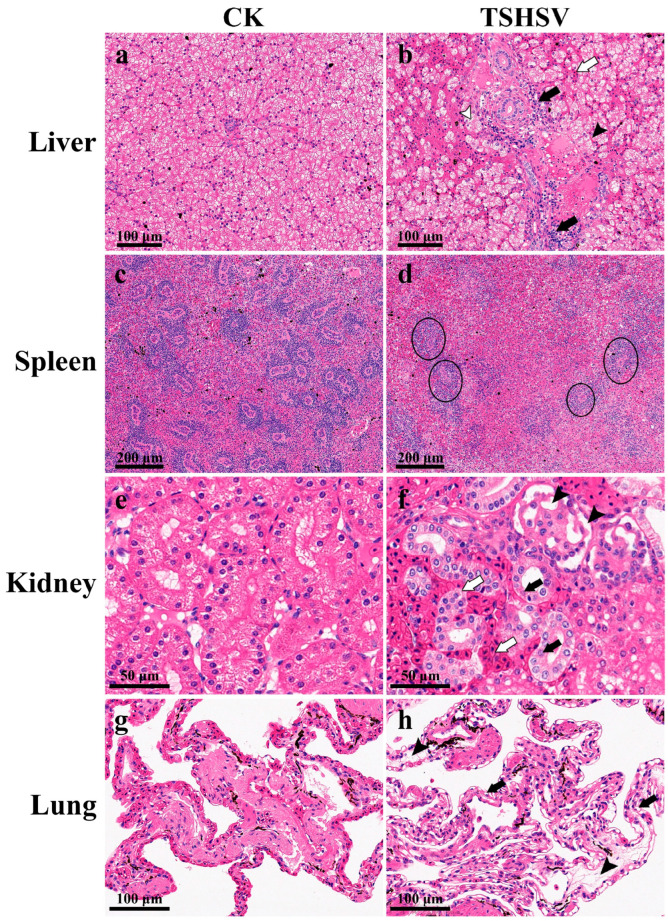
The pathological sections of TSHSV-infected tissues. CK represents the normal group, and TSHSV represents the TSHSV-infected group. (**a**) The normal liver. (**b**) 
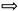
: Hemocyte infiltration, 

: inflammatory cell infiltration, ➤: pyknotic nuclei and cytoplasmic vacuolation, 
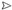
: dotted necrosis and bridging necrosis between cells. (**c**) The normal spleen. (**d**) Atrophy and necrosis in the lymph nodes. (**e**) The normal liver. (**f**) 

:Pyknotic nuclei, nuclear fragmentation, and karyolytic nuclei, 
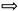
: tubules were filled with blood cells and tubules atrophied, ➤: apoptosis forms vacuoles in the glomerulus. (**g**) The normal lung. (**h**) 

: Necrotic and disintegrated alveolar cells, ➤: disintegrated elastic fibers in alveolar walls.

**Figure 3 animals-13-03665-f003:**
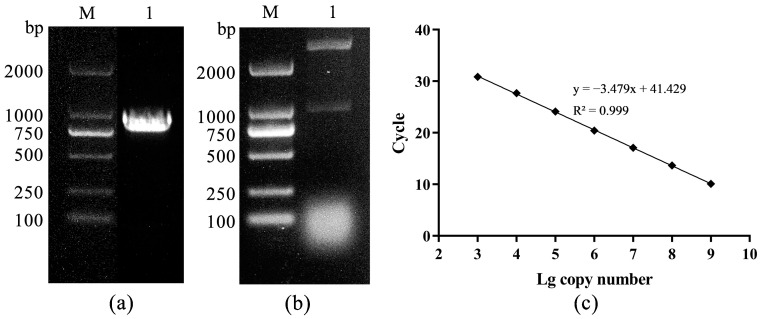
(**a**) The PCR amplification product, M: DL2000 DNA Marker, 1: PCR product, (**b**) the enzyme digestion identification, M: DL2000 DNA Marker, 1: double enzyme digestion, (**c**) the standard curve.

**Figure 4 animals-13-03665-f004:**
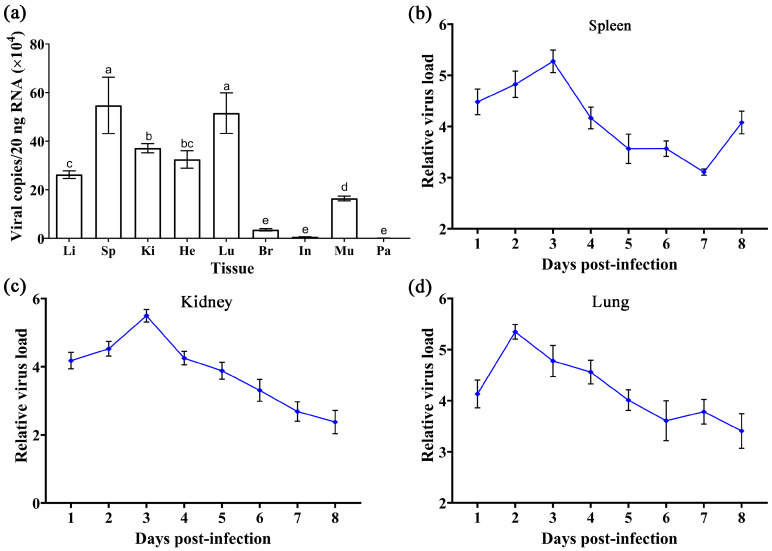
(**a**) The distribution of TSHSV in tissues of naturally diseased turtles. Data are presented as mean ± SD from three biological replicates. The horizontal axis from left to right represents the liver, spleen, kidney, heart, lung, brain, intestine, muscle, and parotid gland. Columns with different letters indicate significant differences (*p* < 0.05), while the same letters indicate no significant difference (*p* > 0.05). (**b**–**d**) The trend of viral replication in the spleen, kidney, and lungs in artificially infected turtles. Results are shown as mean ± SD of three biological replicates.

**Figure 5 animals-13-03665-f005:**
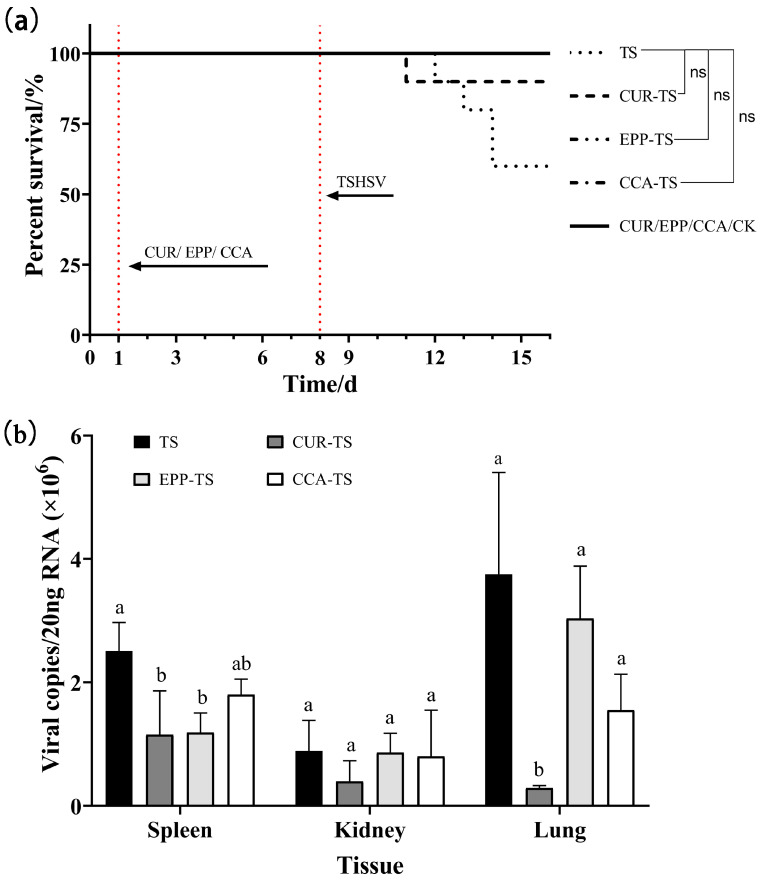
(**a**) The survival curves of *T. sinensis* under three different drugs. (**b**) The inhibitory effect of three different drugs on viral load. For the same tissue, columns with different letters indicate significant differences (*p* < 0.05), while the same letters indicate no significant difference (*p* > 0.05). ns: no significance.

**Figure 6 animals-13-03665-f006:**
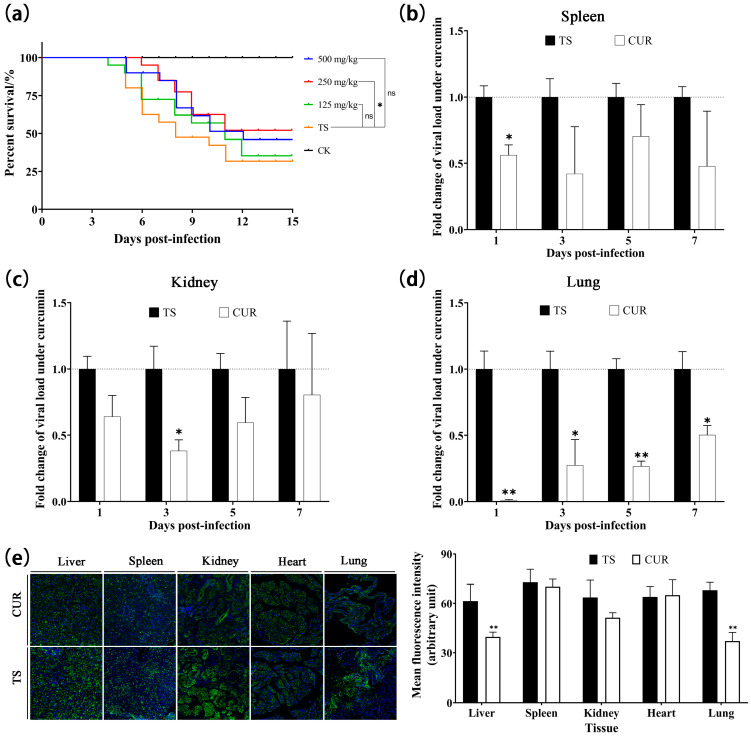
(**a**) Kaplan–Meier survival curves under different concentrations of curcumin. (**b**–**d**) The inhibitory effect of curcumin on viral load in spleen, kidney, and lungs, respectively. (**e**) The inhibitory effect of curcumin on viral protein expression. (* *p* < 0.05, ** *p* < 0.01, ns: no significance).

**Figure 7 animals-13-03665-f007:**
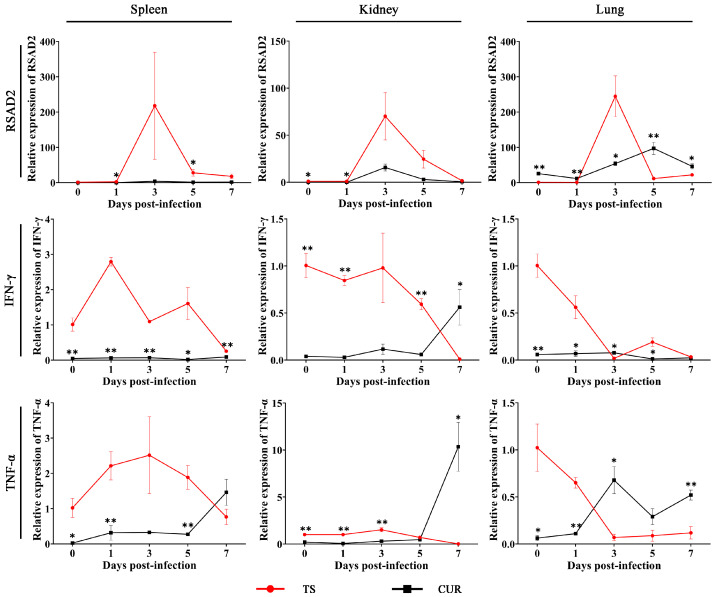
The expression of immune-related genes under curcumin. (* *p* < 0.05, ** *p* < 0.01).

**Table 1 animals-13-03665-t001:** Primers used for TSHSV absolute quantitative method.

Primers	Sequence (5′–3′)	Annealing/°C	Product Length/bp
F1	TCCATCAAGGCTCGTCATGT	60	191
R1	AAGCAGTAGCCATCTCCTGG		
F2	CGGATGATTTTTGGGTACAGATC	58	927
R2	TCAGGGGTTTCCAGATCGG		

**Table 2 animals-13-03665-t002:** Primers of antiviral genes and inflammatory genes in *T. sinensis*.

Primers	Sequence (5′–3′)
*18S rRNA*-F	AAAGGAATTGACGGAAGGGCAC
*18S rRNA*-R	GCTCCACCAACTAAGAACGG
*RSAD2*-F	AGGTATTCCAGTGCCTGCTAAT
*RSAD2*-R	TCCGTCCATGTCTACAGTTCAG
*IFN-γ*-F	CTACTACTCTATCCTGCTCAG
*IFN-γ*-R	GCTTACCTCTGTCCAACTC
*TNF-α*-F	CCATCATCCTCCATCCTTG
*TNF-α*-R	ACGGTCAGTGTGATATGTG

**Table 3 animals-13-03665-t003:** The LD_50_ based on the Reed–Muench method.

Dilutability	Results of Observation			Results of Statistics		
	Death	Survival	Mortality	Death	Survival	Mortality
10^0^	25	5	83.3%	77	5	93.9%
10^−1^	18	12	60.0%	52	17	75.4%
10^−2^	17	13	56.7%	34	30	53.1%
10^−3^	11	19	36.7%	17	49	25.8%
10^−4^	6	24	20.0%	6	73	7.6%

## Data Availability

The data supporting the findings of the study are available from the corresponding author upon reasonable request.

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
