# Peer review of "The Infection Properties of Trionyx sinensis Hemorrhagic Syndrome Virus and the Antiviral Effect of Curcumin In Vivo"

_animals, 2023, doi:10.3390/ani13233665_

Round 1
Reviewer 1 Report
Comments and Suggestions for Authors
The manuscript describes interesting studies with Trionyx sinensis Hemorrhagic 2 Syndrome Virus in cell culture and in vivo. This is an interesting emerging virus that requires greater understanding and treatment and control options. The data from this study contribute to this understanding. The methodology is generally sound, with additional statistics required.
The presentation is not sufficient for publication without extensive English language revision and improved use of specific scientific terminology relating to virology and histopathology.

Extensive revision required. It was difficult to review the science with the current presentation.
Reviewer 2 Report
Comments and Suggestions for Authors
Line 40 - Family is written with a capital letter.
Line 112 - Please insert a link to the developer's website.
Line 122 – 600 s… Is this really true? If so, it's probably better to specify the value in minutes.
Line 152 - Please indicate a specific section.
Line 177 - Please correct. Please also correct any subsequent references in the manuscript.
Line 203 - Please correct. I see several identical figures.
Line 262 - Perhaps it is worth indicating the significance with asterisks as in Figure 6.
Line 293 – Name the graphs b, с and d according to the organ.
Line 417 - Double numbering of the bibliography. Please correct.
I found the results of the work very interesting and useful. Statistical reliability and the number of samples inspire confidence and unambiguity of the results.
1. Supplementary materials, it seems to me, are not needed in this case. These are just raw photos of the gel?
2. Could you clarify please. What is the future of your discovery? Maybe this will be used on turtle farms? Or will a drug be developed?
3. How genetically different are the viruses used in this study from those in the NCBI database? Have you sequenced its genome?
4. The initial titer of infection is 100 (without dilution). Is the initial concentration of the virus known?
5. Not a very good presentation of the data, in my opinion. That is, in the TS group, survival rate fell (the line represented by dots), as did CUR-TS after day 11, but why did EPP and CCA show 100% survival after infection with the virus if curcumin showed the best result? It's a little unclear to me, please explain.
Reviewer 3 Report
Comments and Suggestions for Authors
In this manuscript, cell culture, histopathological analysis, and infection experiment were performed to study the characteristics of TSHSV. In view of the high pathogenicity of TSHSV, the potential antiviral effect of curcumin was studied. The discovery of sensitive cell line and antiviral drug are of great significance in disease prevention. This study has a good innovation and the experiment design is reasonable. However, writing details need to be improved and English grammar needs to be strengthened. The suggestions are as follows:
- Line53. In the introduction, the effects of curcumin on bacteria and parasites are introduced. Then what about curcumin on aquatic animal viral diseases?
- Line 81. Have the cell lines used been reported, or how to obtain them?
- Line 97. Cell experiment showed that TSHSV virus proliferated in cells, so why not use cultured virus for infection experiment?
- Line126. Table 1 seems to need to adjust the horizontal position to fit the manuscript. Please check all the tables and figures.
- Line 140. "n=50, 30 for…and 20 for…". The description of the method is not clear, the definition means that 50 individuals in a group are regrouped?
- Line 263. The description of significance p in the figure note needs to be corrected.
- Line 308. The font size of the horizontal and vertical coordinates in Figure 7 needs to be adjusted.
- Line 319. Please check "but not in lung macrophages". Lung macrophages are not found in the method and result section.
- The sentence in Line 367 regarding the detailed antiviral mechanisms of RSAD2 seems redundant.
- Check the format of references, especially journals and species names.
nothing
Round 2
Reviewer 1 Report
Comments and Suggestions for Authors
Thank you for the response to review. Many of the issues are resolved. The study is interesting and sound. There are some issues relating to presentation of the manuscript that could still be addressed, examples follow:
Some new text is not referenced or given sufficient context (e.g. L62-65) and at line 65-66, the parasite being referred to needs to be provided.
There is still room to improve terminology e.g. L79, invasiveness and infection propensity might actually be captured in the term 'permissive' in context of cultured cells.
L88, the reader needs to know which pathogens they were tested free from, not just the reviewer.
Care with edits, e.g. were is duplicated at L95.
L217, keep all of the caption text relating to c) together.
L229: the term bleeding band is not consistent with terminology used for histopathological descriptions.
L255: terminology. enrichment implies accumulation and concentration, a term that indicates replication is probably intended.
Caption to Fig. 4, still doesn't indicate n and the turtles from which these data were determined.
Fig 5 has unexplained abbreviations e.g. JHS/ZZJ/XDS
Comments on the Quality of English Languagesee previous comment
